# Effect of Heat Treatment on Microstructure and Mechanical Properties of New Cold-Rolled Automotive Steels

**Fei Huang**, **Jian Chen, Zhangqi Ge, Junliang Li and Yongqiang Wang** *

School of Materials Science and Engineering, Anhui University of Technology, No.59, Hudong Road, Hua shan District, Ma anshan 243002, China; maszhjs@126.com (F.H.); 17301526993@163.com (J.C.); gzq9620@163.com (Z.G.); ahut-jonnie@foxmail.com (J.L.)
* Correspondence: yqwang@ahut.edu.cn; Tel.: +86-0555-2311570

**Abstract:** The effect of austenitizing temperature and aging treatment on the microstructure and mechanical properties of two new cold-rolled automotive steel plates (20Mn2Cr and 20Mn2CrNb) was investigated by using isothermal heat treatment, optical microscope, scanning electron microscope, microhardness tester, and tensile testing machine. The results show that as the austenitizing temperature increased, the original austenite grain sizes of both steels increased. The original austenite grain size of 20Mn2CrNb was smaller than that of 20Mn2Cr. The microhardness of 20Mn2CrNb gradually decreased with increasing aging temperature, while the hardness of 20Mn2Cr varied irregularly. The mechanical properties of 20Mn2Cr were better than those of 20Mn2CrNb under the same heat-treatment process. The effect of heat treatment on microstructure and mechanical properties was related to the martensite content, dislocation density, and precipitation of second-phase particles.

**Keywords:** new automotive steel; heat treatment; microstructure; mechanical properties; second-phase particles

## 1. Introduction

The development of automotive steel products with high-strength plasticity is one of the main ways to produce lightweight parts, and is of great significance for energy saving and environmental protection [1–8]. In recent years, there has been a large amount of domestic and international research and development work on high-strength plasticity automotive steel. It has now developed into the third generation, with the main forms being nano-bainite steel, delta transformation-induced plasticity ($\delta$-TRIP) steel, medium manganese steel, quench and partitioning (Q&P) steel, etc. [9–15]. Although each steel grade has its own performance advantages, there are also significant shortcomings.

Nano-bainitic steel is subjected to a low-temperature bainite phase transition region (125–300 °C) for a long isothermal time (several or even tens of days), which is not conducive to large-scale application in the actual production process, and it has a high carbon equivalent and poor welding performance. In order to solve the welding problem, medium- and low-carbon bainitic steels have many production processes and are difficult to implement in practical industrial production and application [16,17].

$\delta$-TRIP steel has a high Al content, which poses problems for smelting and casting, such as clogging of spouts, and its tensile strength is low. Medium manganese steel in the 1000 MPa class with a strength plasticity product of 30 GPa% or more requires a very long annealing time, which is a major problem in actual industrial production in terms of energy consumption, efficiency, and cost. In addition, the high Mn content in medium manganese steel is prone to segregation, the high C content

is not conducive to welding, the Al element will lead to continuous casting problems and cracking in the hot rolling process, and the addition of Si will cause difficulties with hot galvanizing and other application problems. Therefore, there is a considerable gap between actual industrial production and the application of middle manganese steels, especially those with ultra-high-strength [18,19].

For low-carbon or low-alloy content Q&P steel, its tensile strength is generally less than 1500 MPa and maximum elongation is about 15%; increasing the carbon content or adding Nb, Cr, and other elements can further improve strength, but the elongation does not show obvious improvement. In the actual production process, for one-step production, it is often required to precisely control the cold-rolling annealing process, which is bound to need the support of a special high-strength steel production line, and the residual austenite of Q&P steel is generally only about 10% [20], less than that required by high plasticity. If a two-step production method is used, it will obviously lead to problems such as decreased productivity and increased cost and energy consumption in actual industrial production.

At present, the composition design and process optimization of typical third-generation automotive steels are centered on improving plasticity by regulating the residual austenite in the steel and fully utilizing the TRIP effect. While good performance can be obtained with this option, it faces two problems: first, this kind of steel cannot be used in hot forming and is difficult to use in cold forming at higher strength; second, the alloy content is higher, the process is more complex, and the cost is higher. Although third-generation automotive steel has undergone more than 10 years of research and development, most of this steel still does not meet the basic conditions for industrial production and application.

For the promotion and application of third-generation automotive steel, a new reinforced plastic mechanism under ultra-high-strength conditions needs to be put forward through research of key scientific issues and breakthroughs in key technologies. Our research group proposed the design of high-strength plastification steel (PRM) from the interaction of second-phase particles (precipitation) and dislocations (zero- and one-dimensional) and microstructure submicron (two-dimensional) and multiphase (three-dimensional) refinement, breaking through the technical bottleneck of the single mechanism of residual austenite plastification. Based on this idea, new low-alloy steels for flexible applications with high strength plasticity for automobiles were designed and developed, and preliminary cold-rolled steel sheets and hot-formed parts with relatively good mechanical properties were obtained [21–23].

In order to investigate the strong plasticizing mechanism and further improve the properties, the influence of heat treatment on the microstructure, and the mechanical properties, the effects of Nb on the microstructure and mechanical properties were studied to provide theoretical guidance for obtaining better properties.

## 2. Materials and Methods

For the experiment, C-Mn-Cr series low-alloy steels were used, and their main chemical composition is shown in Table 1. The steels were smelted by a 130 kg vacuum induction furnace (Multi-VIM-200, SY-VAC Inc., Shenyang, China), cast into ingots, and forged into 250 mm × 150 mm × 40 mm billets at 1200 °C, then hot rolled into 4.4 mm thick steel plates. After the hot rolling mill, they were kept at 650 °C for 1 h and cooled to room temperature, and finally, the hot rolled plates were pickled and cold-rolled for 10 passes to 2 mm thickness.

**Table 1.** Chemical composition of new cold-rolled steels (wt%).

| Composition | C | Mn | Cr | Nb | Si | Ti | S | P | N | Fe |
|---|---|---|---|---|---|---|---|---|---|---|
| 20Mn2Cr | 0.21 | 1.69 | 1.30 | 0.009 | 0.04 | 0.002 | 0.005 | 0.007 | 0.0098 | balance |
| 20Mn2CrNb | 0.21 | 1.66 | 1.21 | 0.034 | 0.05 | 0.010 | 0.003 | 0.007 | 0.0120 | balance |

The samples were taken from the billets and machined into several solid specimens of Φ4 mm × 10 mm and hollow specimens of Φ4 mm × 10 mm × Φ0.5 mm for use in the continuous cooling transform (CCT) curve measurement by thermal dilatometer (DIL805A/D, BAEHR Inc., HÜllhorst, Germany). The specimens were heated to 980 °C at a heating rate of 10 °C/s, kept hot for 3 min, and then cooled to room temperature at cooling rates of 0.2, 0.5, 1, 2, 5, 10, 20, 30, 40, 50, and 60 °C/s. Solid specimens were used when the cooling rate was 20 °C/s or less, and hollow specimens were used when the cooling rate was more than 20 °C/s.

Heat treatment with different parameters was used for cold-rolled plates, as shown in Figure 1. The heat-treated steel plates were cut into block specimens with dimensions of 15 mm × 15 mm × 2 mm and tensile specimens 30 mm in gauge length and 10 mm × 2 mm in cross-section by wire-cutting, as shown in Figure 2. Heat-treated block specimens were ground to 2000# by abrasive paper, then polished to 0.5 μm with diamond abrasive paste, cleaned by anhydrous ethanol and dried with cold air, and etched by 4% nitric acid alcohol or saturated bitter acid solution. Microhardness was tested with the Vickers hardness tester (TH701, Beijing Time High Technology, Beijing, China) based on Chinese standards (GB/T 4340.1-2009), and tensile tests were carried out on a tensile testing machine (Zwick/Roell Z050, ZwickRoellInc, Ulm, Germany) at a strain rate of 10 MPa/s according to Chinese standards (GB/T 228.1-2010 Method B). The tensile direction was parallel to the rolling direction of the specimen.

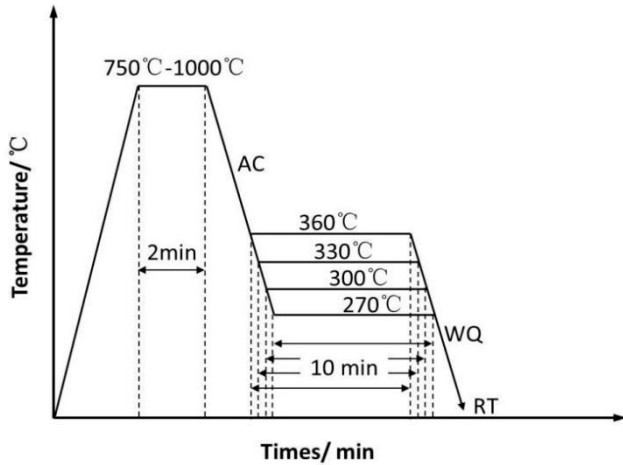

**Figure 1.** Schematic diagram of heat treatment procedure.

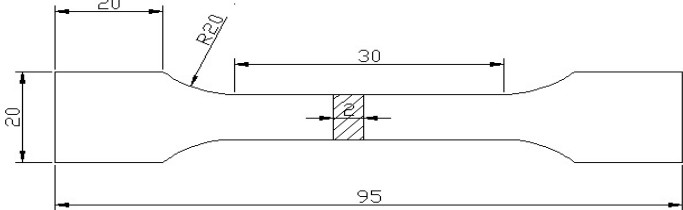

**Figure 2.** Dimensional drawing of tensile specimens (dimensions in mm).

## 3. Result and Discussion

*3.1. Microstructure Transformation Curves of New Cold-Rolled Automotive Steels*

The CCT curves for the two steels by experimental measurements are shown in Figure 3. It can be seen from the figure that the phase transition temperatures ($A_{c3}$ and $A_{c1}$) are 879 and 727 °C and 881 and 733 °C for 20Mn2Cr and 20Mn2CrNb, respectively. The difference between the two steels is quite small. This is because, on the one hand, they have the same C content and little difference between Mn and Si content; on the other hand, the addition of Nb does not have a significant effect

on the phase transition temperature (although Nb can be solidly soluble to the Fe matrix, but the Nb content is relatively low and the Nb atomic radius is large and the degree of solid solution in the Fe matrix is small). In contrast, some differences in the CCT diagrams are observed. These may relate to differences in previous austenite grain size. The previous austenite grain of 20Mn2CrNb steel is finer than that of 20Mn2Cr due to the precipitation of second-phase particles, such as Nb(C,N) and Ti (C,N), which can hinder the growth of austenite grains during heating. The fine grains bring more grain boundaries in 20Mn2CrNb steel. There are more prior sites for nucleation of ferrite and pearlite. Thus, ferrite–pearlite transformation of 20Mn2CrNb steel is promoted and martensite transformation is delayed. Consequently, it can be seen that there is a higher $A_{c1}$ temperature of 20Mn2CrNb steel and no Nb (20Mn2Cr) steel shows higher hardenability revealing.

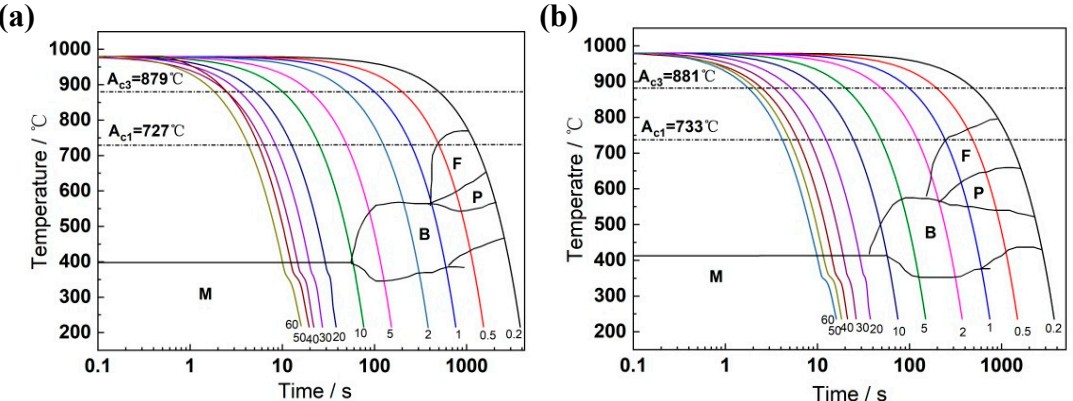

**Figure 3.** Continuous cooling transform (CCT) curves of (**a**) 20Mn2Cr and (**b**) 20Mn2CrNb steels.

*3.2. Microstructure of Cold-Rolled Sheet and Aged Samples*

The microstructure of 20Mn2Cr and 20Mn2CrNb steel samples in the cold-rolled state is shown in Figure 4. It can be seen that the microstructure of both steels consists of ferrite and pearlite. There is little difference in ferrite grain size between the two. Nb and Ti are strong carbide-forming elements; second-phase particles such as Nb(C,N) and Ti(C,N) can be precipitated in the hot-rolling process (in austenite) to hinder the growth of austenite grains, resulting in a reduction in the size of ferrite grains after the phase change [24,25]. However, because of the slow cooling rate during the hot rolling test, the grains have sufficient time to grow, and the phase deformation nucleation rate is low, the phase particle precipitation nucleation rate is also low and the size is larger. Therefore, the effect of adding Nb and Ti on grain size refinement of the steel sample is not significant.

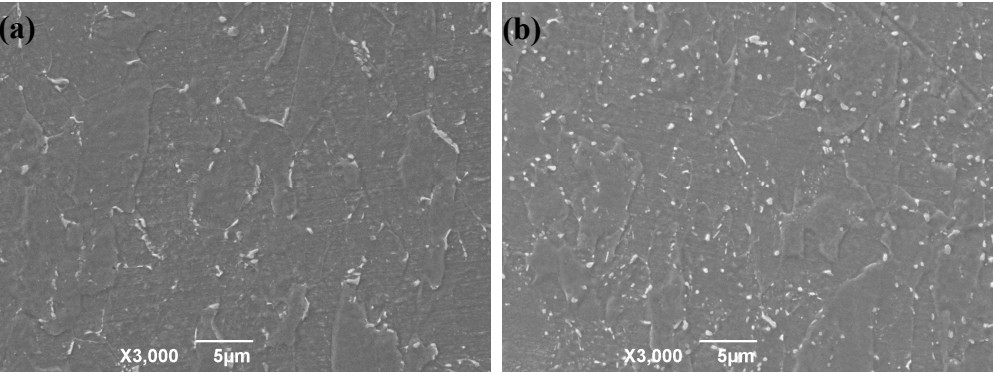

**Figure 4.** Metallurgical microstructure of two cold-rolled plates: (**a**) 20Mn2Cr and (**b**) 20Mn2CrNb.

Figure 5 shows the Scanning Electron Microscope (SEM) microstructure of the two steels after being held at austenitizing temperatures of 800, 900, and 1000 °C for 2 min and then aged at 300 °C

for 10 min. It can be seen that the higher the austenitizing temperature, the more pronounced the martensitic lath characteristics and the larger the original austenite grain size. In addition, the original austenite grain size of 20Mn2CrNb steel at 900 and 1000 °C austenitizing temperature (6.22 μm and 23.14 μm) was slightly smaller than that of the 20Mn2Cr steel (7.23 μm and 27.03 μm), as shown in Figure 6. The CCT curves show that 800 °C is the two-phase zone temperature (Figure 2), so this temperature holding treatment is partially austenitic, and the microstructure of samples consisted of ferrite and martensite. The higher the temperature, the faster the rate of atomic diffusion, the faster the rate of grain boundary migration, and the larger the grain size, as shown in Figure 6. Although there is no significant difference in the grain size of the two steel samples of cold-rolled plate, because Nb is a strong carbide-forming element, the carbide or carbon nitride will not be completely decomposed in Nb-containing steel, so the growth of austenite grain can be hindered to some extent by the residual second phase at a higher austenitizing temperature (900 or 1000 °C).

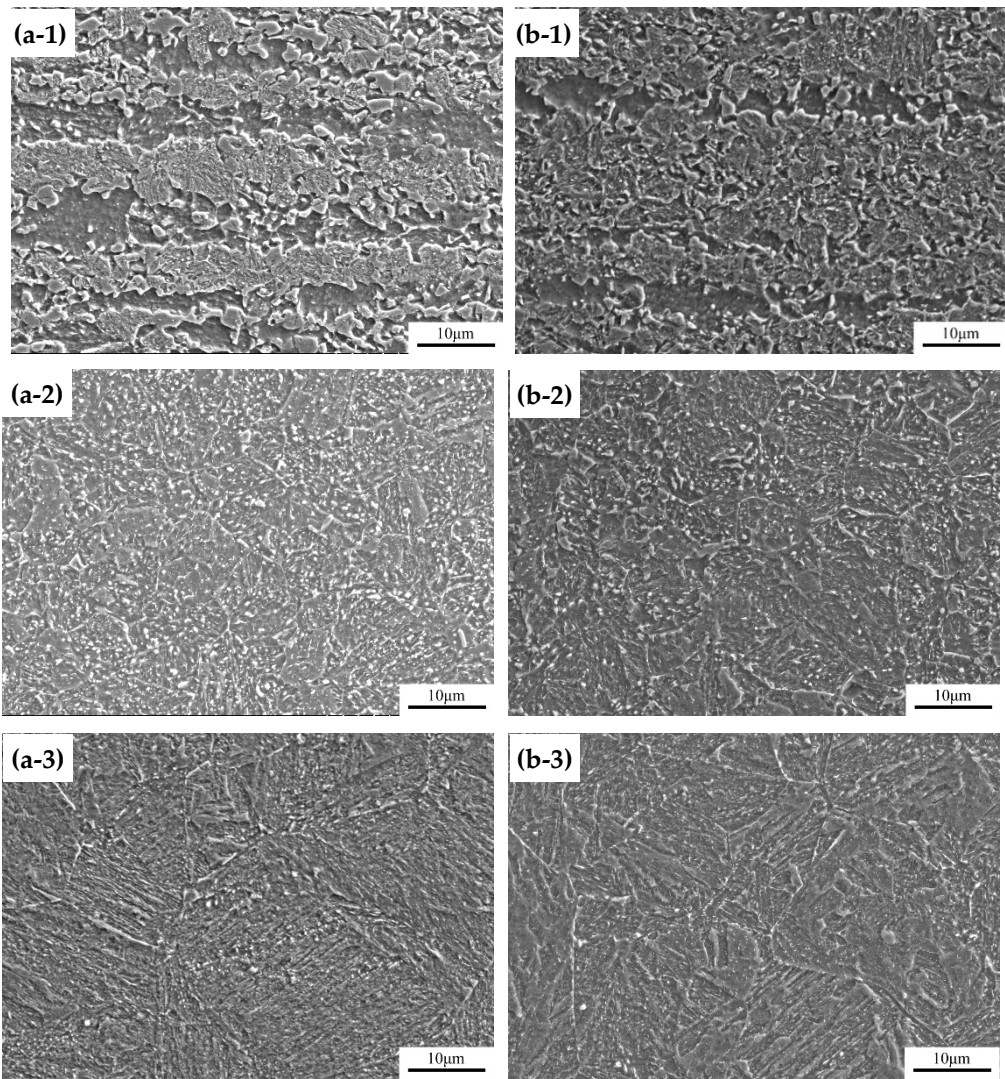

**Figure 5.** Scanning Electron Microscope (SEM) morphology of (**a**) 20Mn2Cr and (**b**) 20Mn2CrNb specimens aged at 300 °C for 10 min after being austenitized at (**1**) 800 °C, (**2**) 900 °C, and (**3**) 1000 °C.

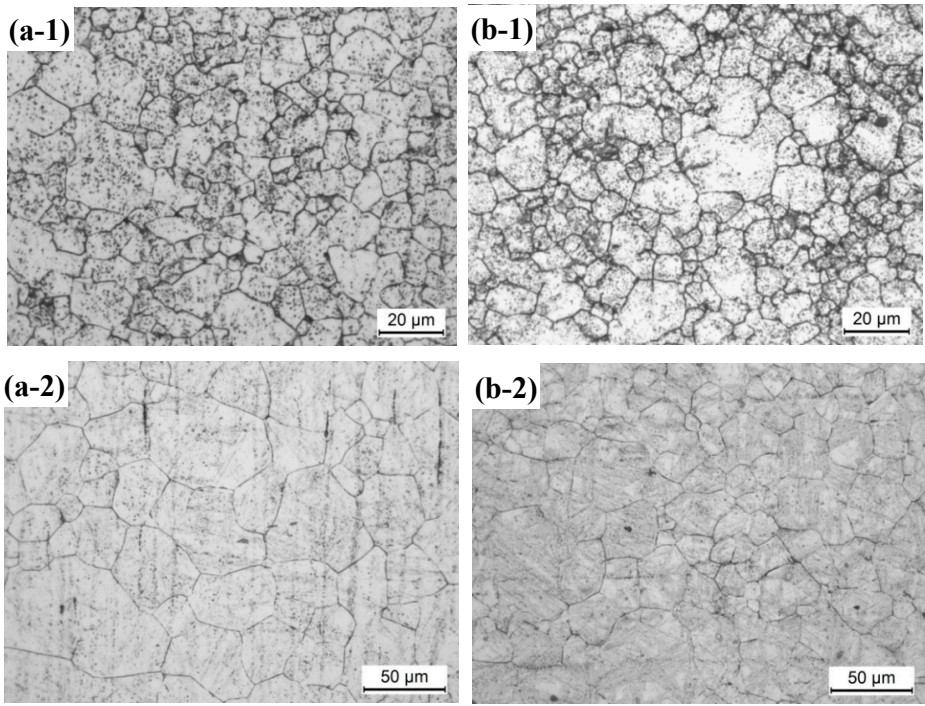

**Figure 6.** Primary austenite microstructure of (**a**) 20Mn2Cr and (**b**) 20Mn2CrNb specimens aged at 300 °C for 10 min after being austenitized at (**1**) 900 °C and (**2**) 1000 °C.

Figure 7 shows the SEM morphology of two steel samples (20Mn2Cr and 20Mn2CrNb) after heat treatment at the same austenitizing temperature (900 °C) and different aging temperatures (300 and 360 °C). It can be seen that as the aging temperature increased, the tempered martensite microstructure of both steels increased and became fine. The reason for this is the higher the aging temperature, the faster the atomic diffusion rate, the more the martensite decomposition, and the higher the amount of Cr, Nb carbide or carbonitrides.

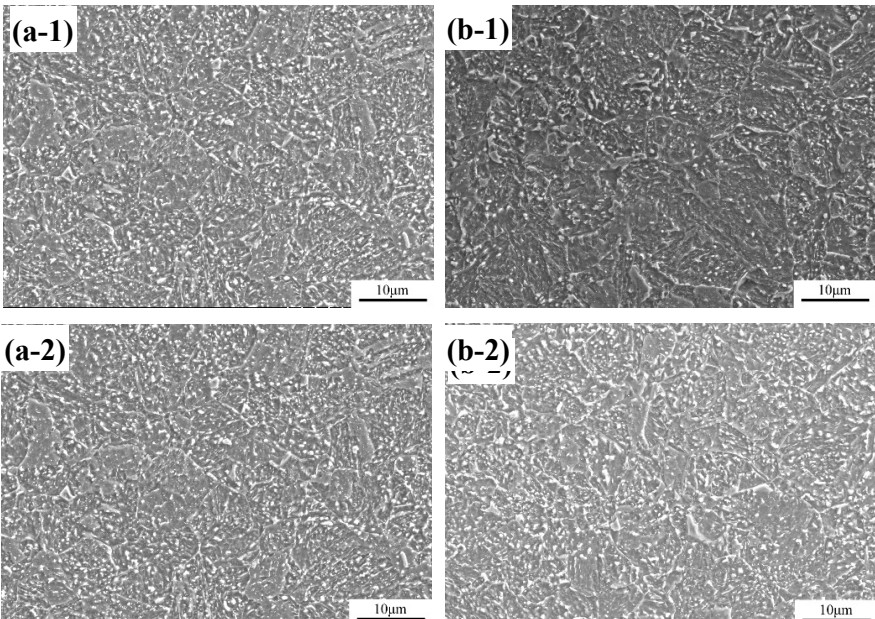

**Figure 7.** SEM morphology of (**a**) 20Mn2Cr and (**b**) 20Mn2CrNb specimens austenitized at 900 °C for 2 min aged at (**1**) 300 °C and (**2**) 360 °C for 10 min.

*3.3. Effect of Aging Temperature on Mechanical Properties of Auto Panels*

3.3.1. Effect of Aging Temperature on Tensile Properties

The room-temperature tensile properties of the two steel samples treated by holding at 850 °C for 2 min then air-cooling to 300 °C for 10 min are shown in Table 2. It can be seen that the tensile strength of the 20Mn2Cr sample after simple heat treatment was greater than 900 MPa, the elongation was nearly 20%, and the mechanical properties were close to the comprehensive properties of QP980. The yield strength (882 MPa), tensile strength (907 MPa), and elongation (18.1%) of the steel sample containing Nb are all lower than those of the steel sample without Nb. Under this heat treatment condition, the addition of Nb is not beneficial to the mechanical properties of the steel. This may be due to several reasons. First, Nb carbonitride has a high precipitation temperature and does not fully precipitate in the aging process at low temperatures [26,27], so the precipitation strengthening effect is not significant. Second, Nb is a strong carbide-forming element. Precipitation of NbC or Nb(C,N) consumes atoms, which inhibits the precipitation of Cr carbides in the aging process to some extent, thus reduces the strengthening effect of the second phase. Third, under the conditions of this experiment, Nb has no obvious effect on grain refinement of the microstructure, which contributes little to the improvement of mechanical properties.

**Table 2.** Tensile properties of 20Mn2Cr and 20Mn2CrNb specimens aged at 300 °C for 10 min after austenitized at 850 °C for 2 min.

| Specimen | $R_{p0.2}$/MPa | $R_m$/MPa | $A$/% | $R_m \times A$/GPa% |
|---|---|---|---|---|
| 20Mn2Cr | 920 | 971 | 19.6 | 19.01 |
| 20Mn2CrNb | 882 | 907 | 18.1 | 16.42 |

3.3.2. Effect of Austenitizing and Aging Temperature on Microhardness

The effects of austenitizing and aging temperature on the microhardness of the two steel samples are shown in Figure 8. As the aging temperature increased, the microhardness of both steels increased. When austenitizing temperatures is lower than 900 °C, the hardness of 20Mn2Cr was higher than that of 20Mn2CrNb, while, the value is lower at temperatures above 900 °C, as shown in Figure 8a. The hardness of the 20Mn2CrNb sample gradually decreased as the aging temperature increased, but the hardness of the 20Mn2Cr sample changed irregularly. Under the condition of short holding time, the higher the austenitizing temperature, the higher the atomic diffusion rate, the more solid solution of alloying elements in the steel, the more even the distribution, and the higher the austenitizing degree. Therefore, the amount of residual austenite is less, the martensite content is more after quenching, and the hardness of the sample is high. Although the differences in hardness between both steels austenitizing at above 950 °C are small, it can be seen that the hardness of 20Mn2CrNb steel is higher than that of 20Mn2Cr steel (insert of Figure 8a). There are two main reasons for it. Firstly, the previous austenite grain size of 20Mn2CrNb was smaller than that of 20Mn2Cr at higher austenitizing temperatures (950 and 1000 °C) (Figure 5a-3,b-3) due to the inhibition of grain growth by second phase particles. The finer austenite grains leads to the formation of finer martensite structure, which brings more difficulty of plastic deformation due to the impediment of dislocations sliding by more grain boundaries. Secondly, the number of precipitated particles in 20Mn2CrNb was more than that in 20Mn2Cr steel during the aging process. Precipitated particles can hinder dislocations moving effectively and improve yield strength. As a result, 20Mn2CrNb was harder than 20Mn2Cr. The higher the aging temperature, the lower the dislocation density, although the amount of second-phase particle precipitation will also increase, but the C and Nb bonded first in 20Mn2CrNb, consuming part of the C atoms, so that the amount of carbide precipitation of Cr did not change significantly with increasing temperature [28]. Therefore, the hardness of the 20Mn2CrNb sample decreased with an increasing

aging temperature. The hardness of the 20Mn2Cr sample varied irregularly with aging temperature, which may be related to the more pronounced increase in the number of carbides of Cr.

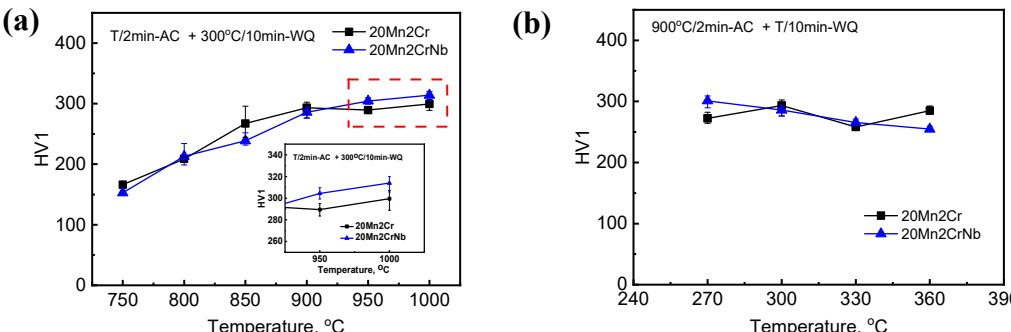

**Figure 8.** Micro-hardness of 20Mn2Cr and 20Mn2CrNb specimens (**a**) austenitized and (**b**) aged at different temperatures.

## 4. Conclusions

(1) The austenitizing temperature had an obvious effect on the microstructure of new cold-rolled automotive steel sheets. As the austenitizing temperature increased, the original austenitic grain size increased and the martensitic slat characteristics became obvious. The original austenite grain size of 20Mn2CrNb with Nb was smaller than that of 20Mn2Cr without Nb at higher austenitizing temperatures.

(2) As the aging temperature increased, the tempered martensite of both steels increased and the microstructure became fine; but for 20Mn2CrNb steel, the higher the aging temperature, the lower the microhardness.

(3) Under the same heat treatment parameters, the tensile mechanical properties of 20Mn2Cr were better than those of 20Mn2CrNb.

**Author Contributions:** Formal analysis, writing—original draft preparation, F.H.; data curation, formal analysis, resources, J.C.; resources, data curation, investigation, Z.G.; validation, visualization, J.L.; methodology, supervision, project administration, writing—review & editing, Y.W. All authors have read and agreed to the published version of the manuscript.

**Funding:** This work was supported by the National Natural Science Foundation of China (grant number 51971003), Natural Science Research Projects of Universities in Anhui Province (grant number KJ2019A0070), Anhui Provincial Science and Technology Program (grant number 18030901085), and Support Program for Outstanding Young Talent in Colleges and Universities of Anhui Province (grant number gxyq2019017).

**Acknowledgments:** The research work in this paper was done under the guidance of Guohui Zhu, and we are grateful to Ding Hanlin of Soochow University and Qiwei Chen of Anhui University of Technology for their help in the research and experimental work.

**Conflicts of Interest:** The authors declare no conflict of interest.

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
