# Peer review of "Effect of Heat Treatment on Microstructure and Mechanical Properties of New Cold-Rolled Automotive Steels"

_metals, doi:10.3390/met10111414_

Round 1

Reviewer 1 Report

Dear author,

I have some comments to your interesting paper.

  • Please check your English first. Some sentences start with a small letter, plural and singular is mixed, words are misse, etc.
  • Is ther any N in the steels? N is not listed in table 1. N can form nitrides, e.g. Ti or Nb. Both elements have a higher content in the Nb variant.
  • Is there any effect of Ti? The Ti content is by a fator of 5 higher in the Nb steel while the factor for Nb is less than 4.
  • What was the thickness of your tensile specimens? It is not mentioned in Fig.2.
  • How did you ensure the high cooling rates? The wall thickness of the hollow samples is 3.5 mm so that there will be a gradient?! In Fig. 9 you've written AC. AC does not follow such fast cooling rates over the whole period. It will be a like a log function.
  • The differences in the CCT curves are 152°C and 148°C for no Nb and Nb, repectively. This is quite small. I would change your statement that they are smaller (you must say for which specimen).
  • The efect of carbonitrides must be explained in 3.2. You refer to Nb(C, N) later but it is not introduced.
  • Can you provide Fig. 4a, b with higher magnifications. I assume this are LOM images? It is not mentioned in the text.
  • It seems to be that there are precipitates at the GB in Fig. 5a3 (bright spots). This seems to be the case in a lesser amount for 5b3. Is this true? Can you comment on that?
  • The adition of Nb (+Ti) seems to be detrimental for the mechanical properties (Fig.8). Is there an element which does improve the mech. properties?
  • The effect of austenitization temperature is higher than the one of ageing temperature. The micro-hardness values seem to level at about 300 HV at T>900°C. Why? Is there an optimum? 
  • Regards
  • Reviewer

Author Response

Thank you very much for your consideration.

Our manuscript (Metals-948740) has been extensively revised (resubmitted online). We are pleased to reply the reviewers’ and editor’s comments.

Reviewer: 1

1)    Please check your English first. Some sentences start with a small letter, plural and singular is mixed, words are missed, etc.

Answer:

Thank you for the comment. We have checked English. The imprecise expressions have been revised, as shown in the revised manuscript.

2) Is there any N in the steels? N is not listed in table 1. N can form nitrides, e.g. Ti or Nb. Both elements have a higher content in the Nb variant.

Answer:

Thank you for the comment.There must be some N in steels. However, the studied steelsaresmelted by vacuum induction furnace,N content in the steels is very small because we control it keeping the very low level intentionally (<40 ppm).N is not considered to be the key object in our present work. So N is not listed in table 1.

3)Is there any effect of Ti? The Ti content is by a factor of 5 higher in the Nb steel while the factor for Nb is less than 4.

Answer:

Thank you for this suggestion. The possible effect of Ti has been explained in the revised manuscript.

4)What was the thickness of your tensile specimens? It is not mentioned in Fig.2.

Answer:

The thickness of tensile specimens is 2 mm and supplied in the revised manuscript. Thank you for this comment.

5)How did you ensure the high cooling rates? The wall thickness of the hollow samples is 3.5 mm so that there will be a gradient?! In Fig. 9 you've written AC. AC does not follow such fast cooling rates over the whole period. It will be a like a log function.

Answer:

Thank you for this comment. The high cooling rates, such as 30°C/s, 40°C/s, 50°C/s and 60°C/s, were obtained by the cooling system in the thermal dilatometer when the CCT curves were measured.However, in the process of heat treatment of specimens between austenitized and agedtemperature air cooling (AC) is adopted. High cooling rate is not necessary for the research on the effect of heat treatment on microstructures and properties of specimens.

6)The differences in the CCT curves are 152°C and 148°C for no Nb and Nb, respectively. This is quite small. I would change your statement that they are smaller (you must say for which specimen).

Answer:

Thanks for your suggestion.  The difference in the CCT curves for no Nb and Nb respectively is really quite small. The imprecisestatements are corrected, as shown in the revised manuscript. 

7) The effect of carbonitrides must be explained in 3.2. You refer to Nb(C, N) later but it is not introduced.

Answer:

Thank you for this suggestion.We agree with you. The effect of carbonitrides has been explained in the revised manuscript.

8) Can you provide Fig. 4a, b with higher magnifications. I assume this are LOM images? It is not mentioned in the text.

Answer:

Thank you for this comment.The higher magnification Figures 4 has been provided,as shown in the revised manuscript.

9) It seems to be that there are precipitates at the GB in Fig. 5a3 (bright spots). This seems to be the case in a lesser amount for 5b3. Is this true? Can you comment on that?

Answer:

Thank you for this comment. The bright spots at GB in Fig. 5a3 may be precipitates, for example, Cr3C7 or Cr23C6, because these carbides precipitate in lower ageing temperature. In Fig. 5b3, these bright spots at GB are lesser maybe due to the precipitation of Nb or Ti carbides or carbonitrides. Nb and Ti are strong carbidesand nitrides forming elements which would be combined with C or N.This reduces the content of solid solution C or N and brings the less Cr carbides precipitation in lower temperature during ageing.

10) The addition of Nb (+Ti) seems to be detrimental for the mechanical properties (Fig.8). Is there an element which does improve the mech. properties?

Answer:

Thank you for this comment. In thepresent experimental conditions which simulated industrial production parameters, the effects of Nb (+Ti) did not display. So the properties of specimen without Nb (+Ti) are higher than that of specimen with addition of Nb (+Ti).

11) The effect of austenitization temperature is higher than the one of ageing temperature. The micro-hardness values seem to level at about 300 HV at T>900°C. Why? Is there an optimum? 

Answer:

Thank you for this comment. Yes. There is more significant influence of austenitizaitontemperature below 900℃ on micro-hardness than one of ageing temperature. When temperature is over 900 ℃, the alloy elements in specimens were solid solution sufficiently and the degree of austenization was also enough. The microstructures in specimens austenitizated at temperature above 900 ℃ did not change. So thereis almost no influence of austenitizationtemperature onmirco-hardness of specimens. The micro-hardness values seem to level at about 300 HV at T>900°C.

Others imprecise expressions,grammar errorsand misspellings in manuscript have been corrected too.

All changes for the manuscript have been highlighted in the revised text with yellow.

Thank you very much for your attention.

Best regards,

Sincerely yours,YongqiangWang

Reviewer 2 Report

1.What is the mechanism of non-metallic particles formation in steel, such as TiN, NbN?
2. What is the nitrogen content in the steel? (no data in the table with chemical composition steel 1 and 2)?
3. What is the share of TiN precipitates in steel 1 and 2.
4. Did the Authors do a thermodynamic analysis of the non-metallic particles formation in these steels? What kind of computational models were used to determine this phases (Hillert model?, Adrian, Staffansson)?
5. What is the temperature criterion for the solubility products of phases: NbN, NbC, TiN, TiC.
5. Figures – microstructures. There are not EDS analysis.

Reviewer 3 Report

This paper entitled “Effect of Heat Treatment on Microstructure and Mechanical Properties of New Cold-Rolled Automotive Steels” is evaluated the effect of the austenitizing as well as the aging treatment in the resultant mechanical properties, taking in consideration the martensite content, dislocation density and precipitation of second phases. I think that this work sounds well with a good characterization. However, I think that this work could be improved in the section of mechanical properties (not only with microhardness test). I mean, additional experimental tests related to nanoindentation by AFM analysis or even scratch test should be incorporated in order to amplify the effect of the thermal effect. Finally, and according to the previous comments, I recommend the paper for publication with some major revisions.

Reviewer 4 Report

"Electronic stretching machine" are not used for testing of materials - tensile testing machine is correct. From the stress strain diagram modulus of elasticity about 80GPa can be determinate. It is too low.  It means that the diagram is not valid - values of strain and determined yield strength are not correct. You probably didn't use an axial extensometer - for this type of measurement it is necessary. 

Author Response

Please see the attachment,Thank you very much!

Round 2

Reviewer 1 Report

Dear authoer,

the paper is much better now. I have some comments.

Please check your English, e.g. long time isothermal... I think either exposure is missing or the sentence must be rewritten e.g. long isothermal time,... You mix past and present in one sentence, e.g. chapter 2. Please make it consistent.

What is the N content of the alloys? Where does the N for the Nb-Carbo/Nitride come from.

Can you provide any citation fpr the Nb(C,N) formation.

Regards

Reviewer

Author Response

Thank you very much for your consideration.Please see the attachment. 

Reviewer 3 Report

This paper can be published in the present form.

Author Response

Dear Reviewer 

   Thank you very much for your consideration!!